# A novel broad host range phage phiA85 displays a synergistic effect with antibiotics targeting carbapenem-resistant *Klebsiella pneumoniae*

Zhoufei Li,[1,2] Zheng Fan,[1] Tongtong Fu,[1] Yuchen Chen,[1] Lin Gan,[1] Bing Du,[1] Xiaohu Cui,[1] Guanhua Xue,[1] Yanling Feng,[1] Hanqing Zhao,[1] Jinghua Cui,[1] Chao Yan,[1] Junxia Feng,[1] Ziying Xu,[1] Zihui Yu,[1] Yang Yang,[1] Yuehua Ke,[1] Jing Yuan[1,2]

**ABSTRACT**   Carbapenem-resistant *Klebsiella pneumoniae* (CRKP) poses a major public health threat worldwide due to the limited treatment options available. There is an urgent need to explore new treatment strategies. Although phages are considered to be an effective treatment, most characterized *Klebsiella* phages demonstrate narrow host ranges restricted to specific capsular types (KL types), thereby limiting their clinical applicability. In the present study, we isolated a novel broad host range phage, phiA85, with high lytic activity and biofilm inhibitory efficacy against host strains of 13 different KL types and 12 ST types. Among these host strains, six strains were identified as multidrug-resistant (MDR)-CRKP covering five different KL types. To further optimize phage therapy, this study indicated that phage phiA85 in combination with sublethal concentrations of antibiotics had synergistic effects against CRKP *in vitro* and *in vivo*. Killing curves and biofilm quantification assays revealed that phiA85–imipenem and phiA85–ciprofloxacin combinations exhibited phage–antibiotic synergy (PAS), demonstrating better bactericidal efficacy and reduction in biofilm formation compared to monotherapy. In a mouse pneumonia model, phage phiA85–imipenem combination treatment reduced mortality and alleviated pneumonia without other side effects. Our findings identify a broad host range phage capable of lysing different KL types and ST types of *K. pneumoniae*. Notably, phage phiA85–antibiotic combination offers a promising therapeutic option for the clinical treatment against MDR-CRKP.

**IMPORTANCE**   The widespread prevalence of MDR bacteria has become a critical public health threat with limited therapeutic options. This study identifies a novel broad host range phage phiA85, capable of lysing strains across different KL types of CRKP. This characteristic addresses the limitation of narrow host range of capsular-specific phages, thereby significantly expanding their therapeutic potential against multidrug-resistant bacterial infections. Phage phiA85–antibiotic combinations achieve PAS, enhancing lytic activity and biofilm inhibition *in vitro* and alleviating mouse pneumonia. These findings highlight that phage phiA85–antibiotic combinations are a promising strategy to combat MDR-CRKP infections. Our work provides critical insights into optimizing phage therapy for clinical use against priority pathogens.

**KEYWORDS**   carbapenem-resistant *K. pneumoniae*, multidrug-resistant, broad host range phage, phage–antibiotic combination therapy

Address correspondence to Jing Yuan, yuanjing6216@163.com.

Zhoufei Li, Zheng Fan, and Tongtong Fu contributed equally to this article. The author order was determined in order of increasing seniority.

The authors declare no conflict of interest.

See the funding table on p. 17.

*K*lebsiella pneumoniae is a human pathogen capable of causing nosocomial and community-acquired infections, which is generally categorized as classical *Klebsiella pneumoniae* (cKp) and hypervirulent *Klebsiella pneumoniae* (hvKp) based on their virulence phenotypes and genetic characteristics (1). hvKp is able to cause severe

invasive diseases such as liver abscess, pneumonia, and meningitis in healthy individuals (2, 3). Mucoid regulators (*rmpACD* and *rmpA2*), iron acquisition systems (*iucABCD* and *iutA*), and metabolite transporter *peg-344* are considered typical biomarkers that distinguish hvKp from cKp (4, 5). Compared to hvKp, infections caused by *K. pneumoniae* are predominantly driven by multidrug-resistant (MDR) strains, among which carbapenem-resistant *Klebsiella pneumoniae* (CRKP) poses the most significant therapeutic challenges due to its limited antibiotic susceptibility (6–8). The incidence of CRKP infection is increasing rapidly worldwide, and the mortality rate after infection can be as high as 50% (9). CRKP is considered a "superbug" by the US Centers for Disease Control and Prevention (10). In China, the major prevalent CRKP is the ST11 clonal group, associated with high mortality (11). Carbapenemase genes associated with *K. pneumoniae*, including *Klebsiella pneumoniae* carbapenemase ($bla_{KPC}$), New Delhi metallo-β-lactamase, and oxacillinase ($bla_{OXA}$), have been extensively characterized (9). Among these, KPC is the most widely prevalent class A carbapenemase (12). Ceftazidime-avibactam has been recognized as one of the most efficient drugs against CRKP in recent years (13, 14). However, resistance against ceftazidime-avibactam in CRKP emerged before clinical use (15). The pace of novel antibiotic discovery has been observed to lag substantially behind the emergence rate of antimicrobial-resistant bacterial strains (16). Hence, new therapeutic and prevention strategies for the control of CRKP infections are urgently needed.

Bacteriophages (phages) are emerging as a promising tool in the era of decreasing effective antibiotics (17–19). In recent years, phage therapy has been used to treat diseases caused by MDR pathogens, such as lung infections in transplant patients and bone or joint infections (20, 21). Meanwhile, research on the therapeutic effect of *Klebsiella* phage has been also emerging. Some studies have demonstrated that phages have shown great therapeutic effect on MDR *K. pneumoniae*-induced pneumonia and high alcohol-producing *K. pneumoniae*-induced non-alcoholic fatty liver disease in mice (22–24). The capsular locus (KL) diversity of *K. pneumoniae* restricts the host range of *Klebsiella* phages. Most identified *K. pneumoniae* phages demonstrate bactericidal activity exclusively against hosts with specific KL types, which restricts their clinical applications (15). Few studies have reported broad host range phages targeting multiple KL types of *K. pneumoniae* (25, 26).

In the process of phage therapy, the development of phage-resistant bacteria has also become a matter of concern (27). A variety of studies have reported that phages, in combination with sublethal concentrations of antibiotics, can reduce the probability of drug-resistant bacteria, enhance the bactericidal effect, and inhibit biofilm formation as opposed to monotherapy. The phenomenon is called phage–antibiotic synergy (PAS) (28–30). The application of PAS in clinical cases has been successfully documented. A case of phage–antibiotic interactions curing a 13-year-old girl who developed chronic polymicrobial biofilm infection of a pelvic bone allograft after Ewing's sarcoma resection surgery was reported several years ago (31). In China, a non-active antibiotic and phage combination successfully cured a female patient with a urinary tract infection of drug-resistant *K. pneumoniae* and also inhibited the emergence of phage-resistant mutants (32). Hence, phage–antibiotic combination becomes an optimized phage therapy for treating drug-resistant bacteria infections.

In this study, we isolated a novel *Klebsiella* phage, phiA85, which had a broad host range against *K. pneumoniae* of different ST types and KL types, which showed general lytic activity on CRKP. We next evaluated the combinatorial efficacy of phage phiA85 with sublethal-dose antibiotics against CRKP. This synergistic regimen demonstrated significantly enhanced bactericidal activity in killing curves and biofilm inhibition assay and effective therapeutic outcomes in mouse pneumonia models compared to phage monotherapy. Our findings demonstrate that the phage phiA85–antibiotic combination exhibits a promising therapeutic potential against CRKP infections.

## RESULTS

### A85: a CRKP with typical characteristics of hvKp

A strain of *K. pneumoniae* named A85 was isolated from the bronchoalveolar lavage fluid of a clinical pneumonia patient, which belonged to ST11 and carried four plasmids. One of the plasmids (A85-p2) contained the *K. pneumoniae* carbapenemase gene (*bla*$_{KPC-2}$) and several β-lactamase genes (*bla*$_{SHV-12}$ and *bla*$_{CTX-M-65}$) (Fig. 1A). Other drug resistance genes such as *SHV-11* and quinolone resistance genes (*gyrA* and *parC*)

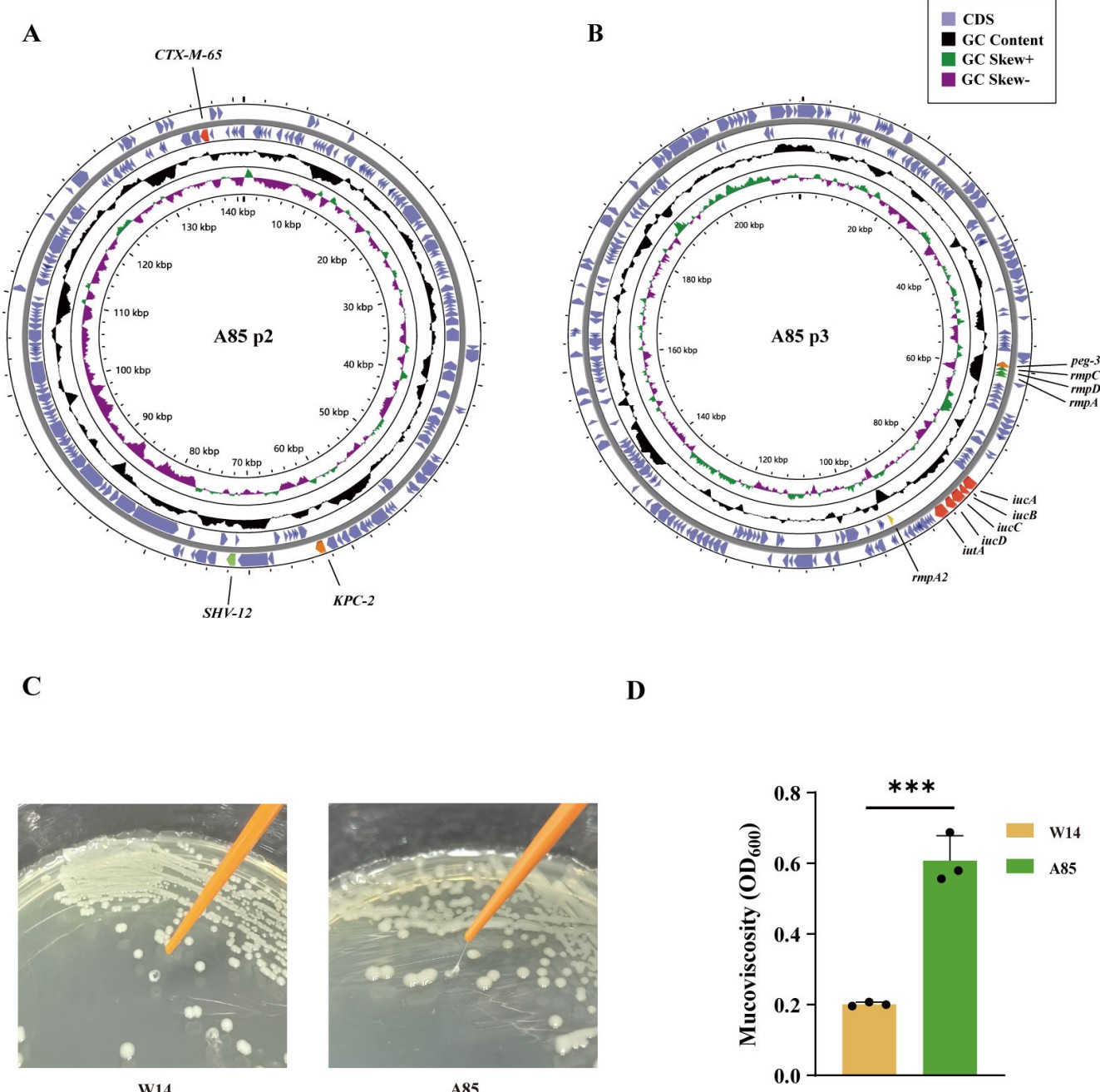

**FIG 1** The mucoid and genomic features of a carbapenem-resistant *Klebsiella pneumoniae* (CRKP) A85. (A and B) The genomic map of A85 plasmid 2 (A85 p2) and A85 plasmid 3 (A85 p3). (C) String test of W14 and CRKP A85 colonies. (D) Mucoviscosity test between A85 and W14 *K. pneumoniae*. Data represent mean ± SD (B, *n* = 3). Statistical significance was determined using Student's *t*-test. ***P < 0.001; CDS, coding domain sequence; GC, guanine and cytosine; OD$_{600}$, optical density with a wavelength of 600 nm.

were found in the chromosome (Fig. S1A). The minimum inhibitory concentration (MIC) test showed that A85 was found to be an MDR-CRKP that was resistant to imipenem, ciprofloxacin, levofloxacin, ceftazidime, and azithromycin (Table 1). Notably, A85 concurrently manifested the characteristics of hvKp. We found a 217,900 bp plasmid A85 p3 carrying hypermucoviscosity genes including mucoid regulators (*rmpACD* and *rmpA2*), iron acquisition systems (*iucABCD* and *iutA*), and metabolite transporter *peg-344* (Fig. 1B). Compared to a negative-string test cKp strain W14, A85 showed a positive-string test on Luria–Bertani (LB) agar plate (Fig. 1C). In the mucoviscosity test (low-speed centrifugation test), the supernatant of A85 exhibited turbidity in comparison with W14, which was a classical *Klebsiella pneumoniae* strain with capsular type KL30, isolated from the patient's intestine (Fig. 1D). The tests showed that A85 exhibited a hypermucoviscosity phenotype.

## Phage phiA85: a broad host range phage against *K. pneumoniae* of different ST types and KL types

A lytic phage named phiA85 was isolated from hospital sewage against A85. The phage formed clear plaques in double-layer agar plates with a diameter of about 2.0 mm (Fig. 2A). The transmission electron microscopy (TEM) analysis clearly showed that the phage phiA85 had a polyhedral head of about 64–111 nm in diameter and a contractile tail of about 118 nm (Fig. 2B), belonging to *Caudoviricetes* family order of dsDNA viruses. The optimal multiplicity of infection (MOI) of phage phiA85 was 1:1,000 (Fig. 2C). The one-step growth curve showed that the latent period was 50 min, and the burst time was 130 min (Fig. 2D). The phage remained stable in environments between 25°C and 50°C. A significant decrease was observed from 60°C to 80°C, and no plaque was observed at 90°C and 100°C (Fig. 2E). Phage phiA85 maintained a good activity in the pH range of 6–9 (Fig. 2F). The genome sequencing results indicated that phage phiA85 contained 167,232 bp in length with 39.57% guanine and cytosine content. A total of 290 coding domain sequences (CDSs) were predicted. Based on the putative functions of the phage-encoded genes, the genome was divided into four functional domains: structure, DNA packing, DNA replication, and host lysis. A total of 176 gene products were similar to proteins with known functions, while the other 114 genes were noted as hypothetical proteins. The genomic data showed that phiA85 carries no virulence or antibiotic resistance that was suitable for phage therapy (Fig. 2G). Based on lifestyle prediction analysis, this phage is classified as a virulent phage (Table S2). Phylogenetic tree analysis showed that phage phiA85 was clustered with *Klebsiella* phage KMI13 on a separate branch, which belongs to *Caudoviricetes* family, genus *Straboviridae* (Fig. 2H), suggesting that they are close relatives. To determine the host range of phage phiA85, we selected 56 strains, including different KL types and other gram-negative bacteria, and 22 isolates were verified to be lysed by phage phiA85 (Table S1). Phage phiA85 was

**TABLE 1** Antimicrobial susceptibility of *K. pneumoniae*[a,b]

| Antibiotics | Minimum inhibitory concentrations (MIC) (µg/mL) | | | | | |
| --- | --- | --- | --- | --- | --- | --- |
| | B35 | B38 | B58 | B59 | A85 | B92 |
| Azithromycin | 128$^R$ | 128$^R$ | 128$^R$ | 128$^R$ | 128$^R$ | 128$^R$ |
| Ciprofloxacin | >64$^R$ | >64$^R$ | >64$^R$ | >64$^R$ | 64$^R$ | 64$^R$ |
| Levofloxacin | >64$^R$ | >64$^R$ | 64$^R$ | 32$^R$ | 32$^R$ | 32$^R$ |
| Amikacin | >64$^R$ | >64$^R$ | >64$^R$ | >64$^R$ | 4$^S$ | 4$^S$ |
| Gentamicin | >64$^R$ | >64$^R$ | >64$^R$ | >64$^R$ | 2$^S$ | 2$^S$ |
| Kanamycin | >64$^R$ | >64$^R$ | >64$^R$ | >64$^R$ | 4$^S$ | 8$^S$ |
| Ceftazidime | 32$^R$ | 64$^R$ | 32$^R$ | 64$^R$ | >64$^R$ | >64$^R$ |
| Imipenem | >64$^R$ | >64$^R$ | 64$^R$ | 64$^R$ | 64$^R$ | >64$^R$ |
| Tetracycline | 4$^S$ | 16$^R$ | >64$^R$ | <1$^S$ | 4$^S$ | >64$^R$ |
| Polymyxin B | 8$^R$ | 8$^R$ | 4$^R$ | 4$^R$ | 4$^R$ | 2$^I$ |

[a]I, intermediate; R, resistance; S, susceptible.
[b]Ceftazidime belongs to third-generation cephalosporin.

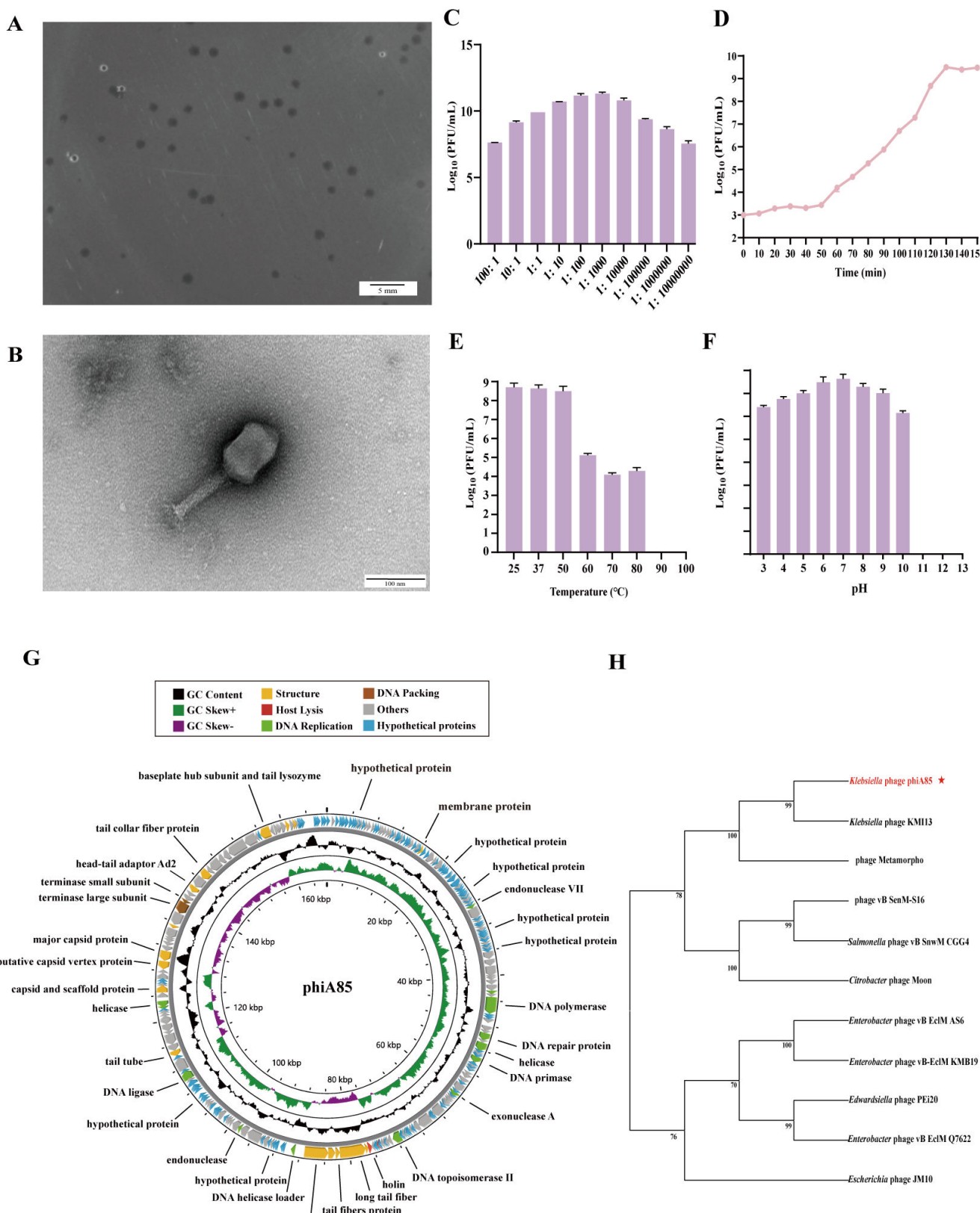

**FIG 2** The morphology, biological characterization, and genomic features of phage phiA85. (A) Phage plaques of phage phiA85. (B) The morphology of phage phiA85 observed by TEM. (C) Optimal multiplicity of infection (MOI) of phage. (D) One-step growth curve. (E and F) Temperature and pH stability of phage. (G) The genomic map of phage phiA85. Data represent mean ± SD (C–F, $n = 3$). (H) Phylogenetic tree constructed from amino acid sequences of the large terminase subunit using the maximum likelihood method.

species-specific, unable to lyse other bacteria species. Then, we sequenced the genomes of 17 host strains of phage phiA85. The results showed that phiA85 had a broad host range against *K. pneumoniae* of 13 different KL types and 12 ST types (Fig. 3A).

## Phage phiA85 demonstrates lytic activity and induces biofilm inhibition against CRKP

Analysis of the resistance genes of 17 host strains revealed that six of them carried carbapenemase genes; B35 carried oxacillinase (*bla*$_{OXA}$); and the other strains carried *K. pneumoniae* carbapenemase (*bla*$_{KPC-2}$). These six strains covered five different KL types (KL24, KL27, KL47, KL109, and KL136). Five of them belonged to the ST11 clone except B92 (Fig. 3A). The antibiotic sensitivity test showed that these strains were found to be MDR-CRKPs, which were resistant to carbapenems (imipenem), fluoroquinolones (ciprofloxacin and levofloxacin), macrolides (azithromycin), third-generation cephalosporin (ceftazidime), and polymyxins (polymyxin B) (Table 1). At the same time, several hypervirulence genes in *K. pneumoniae*, including *rmpACD*, *rmpA2*, and *iucABCD*, were also detected in three strains. By constructing a phylogenetic tree of the phage host strains and some non-host strains, we found that these 17 host strains belong to distinct evolutionary branches (Fig. 3B). To further understand the lytic activity of host bacteria by phage phiA85, killing curves of 17 host strains at MOIs of 0.001, 0.1, and 10.0 showed that phiA85 had different potencies and optimal MOI for different strains. Some bacteria were nearly completely lysed at specific MOIs (Fig. 3C and D; Fig. S2A), while others showed partial lytic activity (Fig. 3E through H; Fig. S3A). Moreover, phage phiA85 could effectively inhibit biofilm formation of these host strains (Fig. S2B). The results indicated that the broad host range phage phiA85 demonstrated lytic activity and induced biofilm inhibition against CRKP against diverse CRKP strains.

## The synergistic effect of phage phiA85–antibiotic combinations *in vitro*

Some laboratory experiments showed that the rates of bacterial resistance evolution decreased when bacteria were treated with phage–antibiotic combinations, compared with phages or antibiotics alone (29, 33). To explore the bactericidal effect of phage phiA85–antibiotic combinations, synergy testing methods (28) with slight modification were applied. The clinically prevalent antibiotics imipenem and ciprofloxacin were selected for combinatorial therapeutic intervention with phage phiA85. We combined imipenem or ciprofloxacin with phage phiA85 in different concentration gradients, and the initial concentration of bacteria in each well was $1 \times 10^7$ CFU/mL. Optical density with a wavelength of 600 nm (OD$_{600}$) in each well was measured after 24 h to calculate the reduction percentage. The heat map results demonstrated that the dose of antibiotics was basically positively correlated with reduction percentage. CRKPs (except phiA85–ciprofloxacin combination in B38) showed nearly complete reduction (>90%) when they were treated with ≥512 µg/mL of imipenem or ciprofloxacin (Fig. 4 and 5). Under combinatorial treatment, there is an almost complete reduction of A85 with concentrations of 64 µg/mL of imipenem, suggesting that the addition of phage completely reduced the concentration by eightfold (Fig. 4A). On the contrary, the addition of antibiotics did not reduce the phage titer needed for effective killing. Furthermore, the phage–antibiotic killing dynamic curves were also applied to assess the effects of phage phiA85–antibiotic combinations. The results indicated that $10^3$ PFU/mL phiA85 and 32 µg/mL imipenem or ciprofloxacin resulted in a complete killing effect compared with either treatment alone in A85; more than 95% reduction was observed in the combination treatment (Fig. 4A and B). In B92, $10^4$ PFU/mL phiA85, in combination with 32 µg/mL ciprofloxacin, also resulted in a desirable bactericidal effect than monotherapy (Fig. 5B). The killing curves of B35 and B38 suggested that phage–antibiotic combinations showed partial synergistic effects, in which phage phiA85 played a major role (Fig. 4C, D, 5C and D).

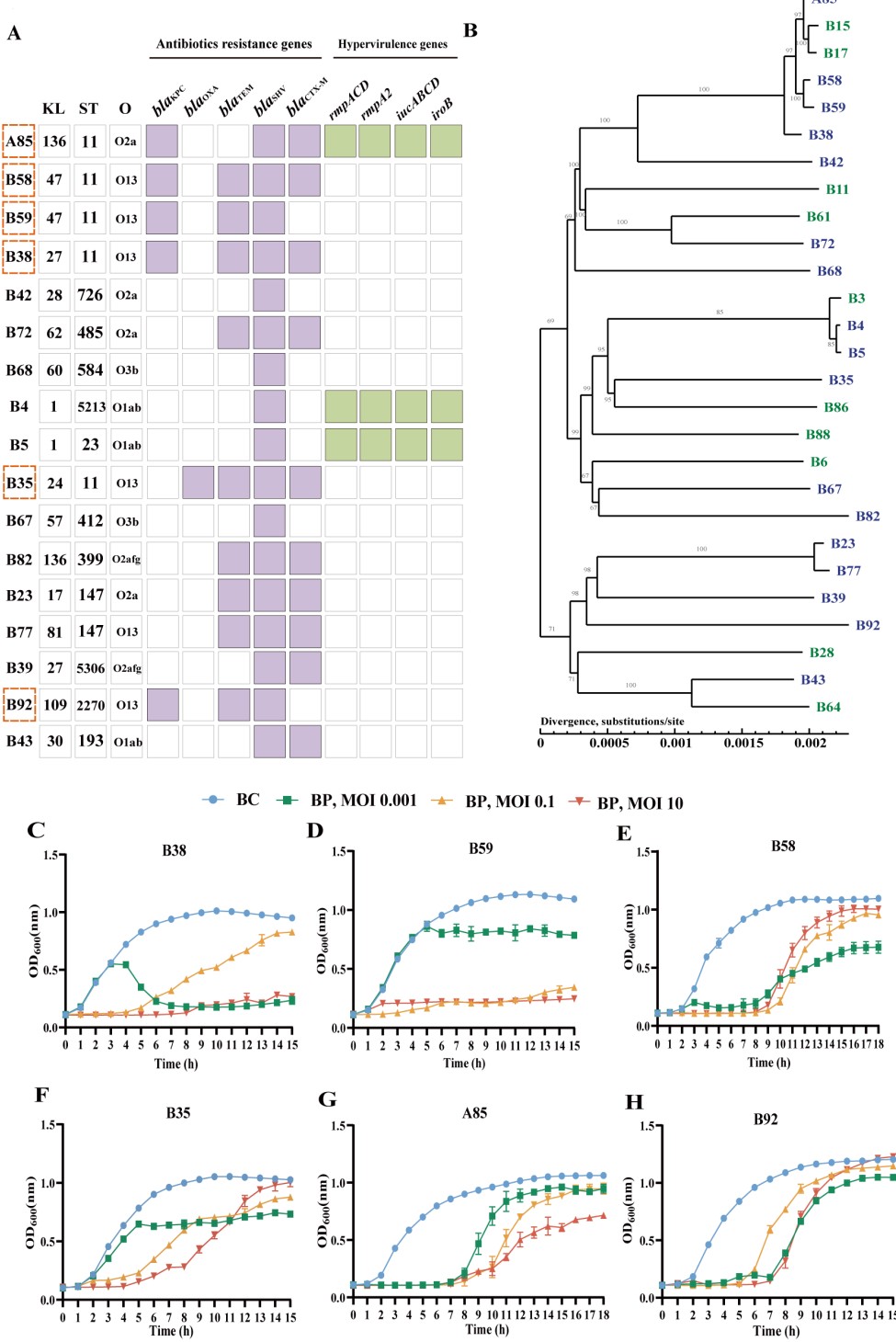

FIG 3 The broad host range of phage phiA85. (A) Heatmap of resistance and hypervirulence genes in 17 phage phiA85 targeted host *K. pneumoniae* strains. Each row represents a host strain, and each column represents the KL types, ST types, O antigens, antibiotic-resistant genes, and hypervirulence genes carried by a host strain. The name of strains with an orange dashed line represents CRKP. (B) Phylogenetic tree of host strains. Dark blue font represents the host strains of phage phiA85, and green font represents *K. pneumoniae* strains that the phage cannot lyse. Phylogenetic tree constructed from corn gene analysis using neighbor-joining method. (C–H) Killing curves of phage phiA85 against eight CRKPs. BC, bacteria control; BP, bacteria and phage mixture in different MOIs. Data represent mean ± SD (C–H, $n = 3$).

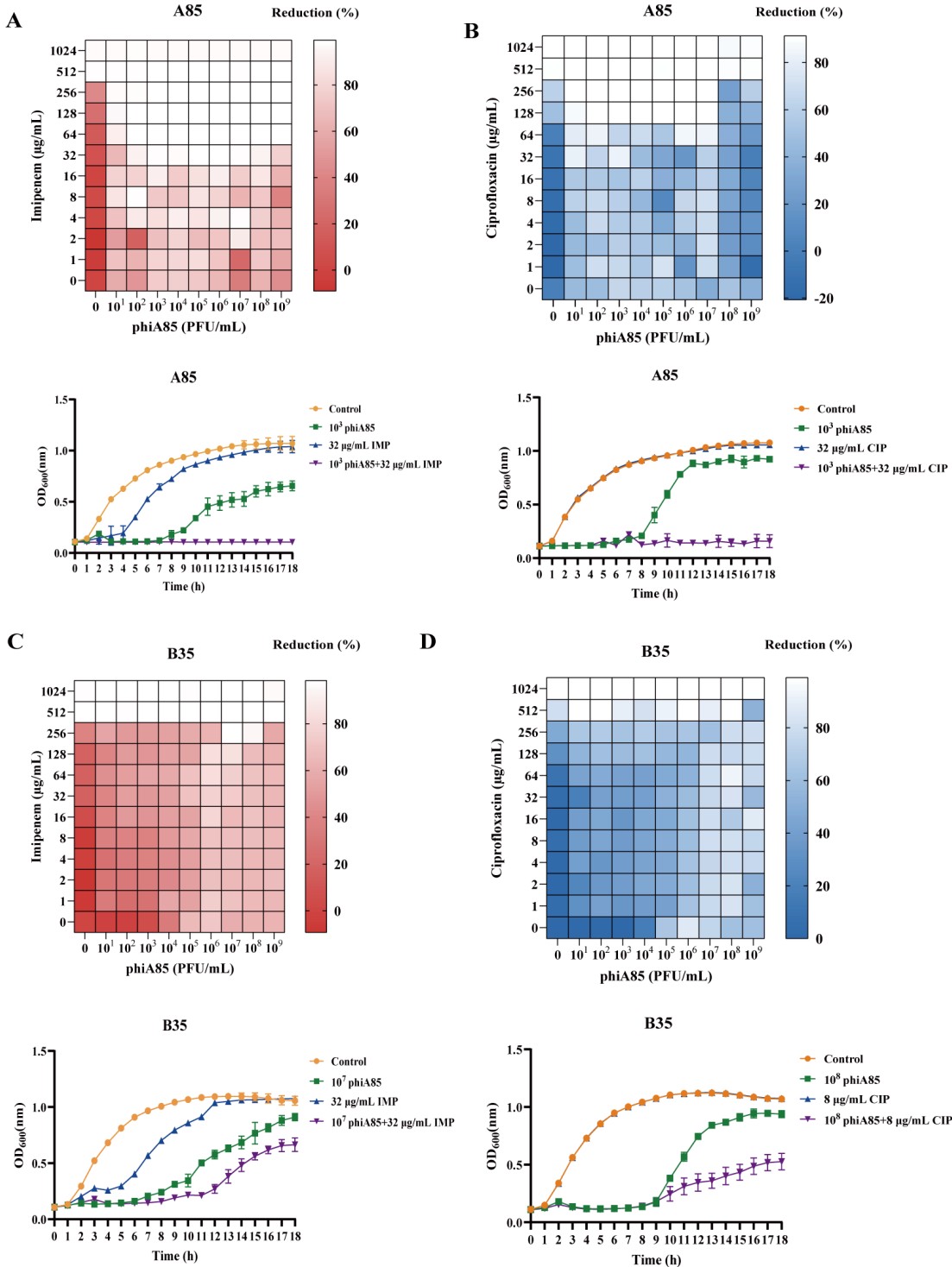

**FIG 4** Heat maps and killing dynamics curves of phage phiA85–antibiotic combinations on A85 and B35. (A and B) Phage phiA85–imipenem or ciprofloxacin combinations on A85. (C and D) Phage phiA85–imipenem or ciprofloxacin combinations on B35. Each row represents concentration gradients of antibiotics (imipenem or ciprofloxacin), and each column represents concentration gradients of phage phiA85. $OD_{600}$ in each well was measured after 24 h to calculate the reduction percentage. Reduction (%) = [($OD_{growthcontrol}$ − $OD_{treatment}$) / $OD_{growthcontrol}$] × 100%. Killing dynamics curves show means ± SDs (*n* = 3). Control means bacteria only. CIP, ciprofloxacin; IMP, imipenem.

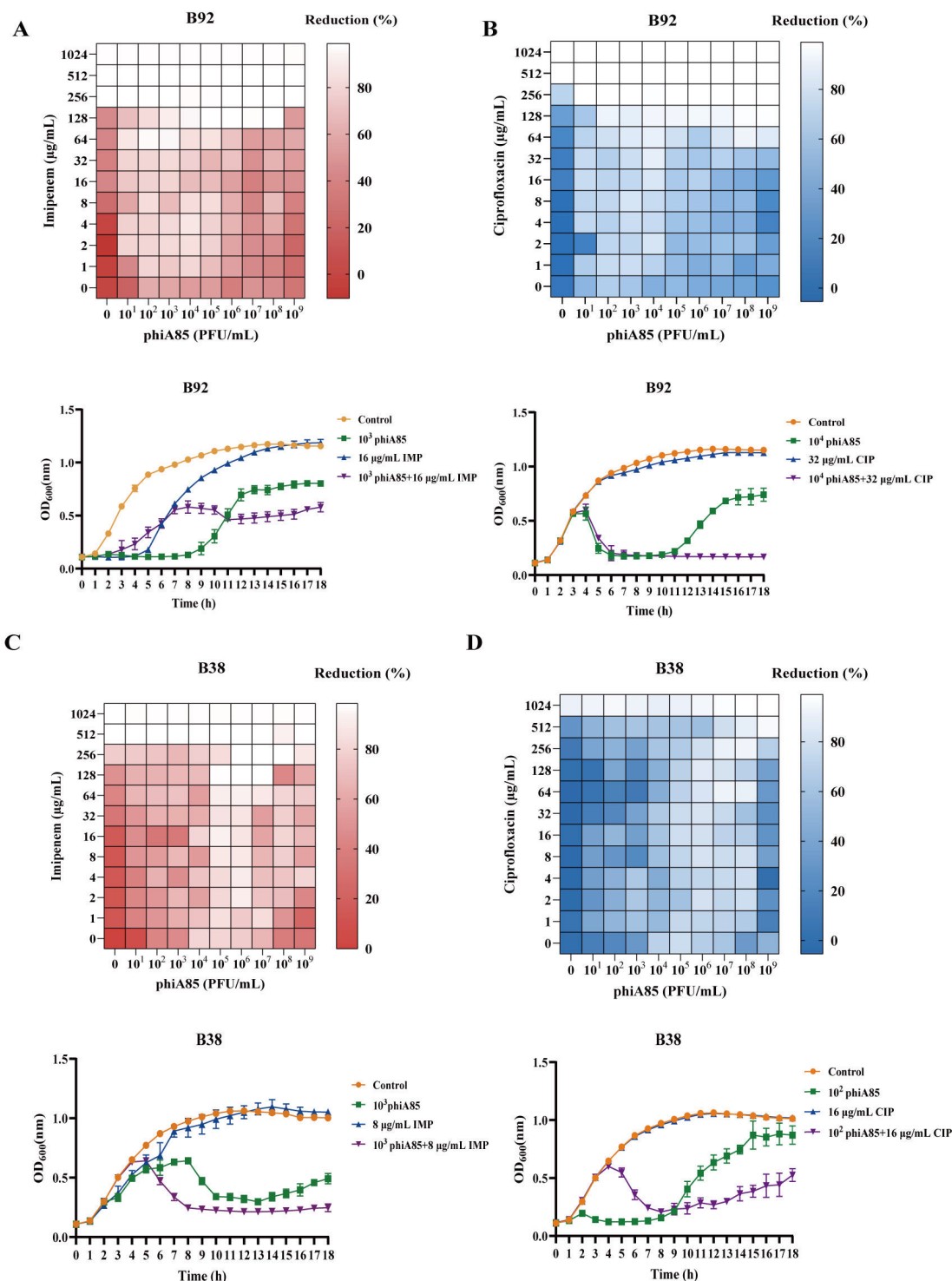

**FIG 5** Heat maps and killing dynamics curves of phage phiA85 and antibiotic combinations on B38 and B92. (A and B) Phage phiA85-imipenem or ciprofloxacin combinations on B92. (C and D) Phage phiA85–imipenem or ciprofloxacin combinations on B38. Each row represents concentration gradients of antibiotics (imipenem or ciprofloxacin), and each column represents concentration gradients of phage phiA85. $OD_{600}$ in each well was measured after 24 h to calculate the reduction percentage. Reduction (%) = [($OD_{growthcontrol}$ − $OD_{treatment}$) / $OD_{growthcontrol}$] × 100%. Killing dynamics curves show means ± SD ($n$ = 3). Control means bacteria only. CIP, ciprofloxacin; IMP, imipenem.

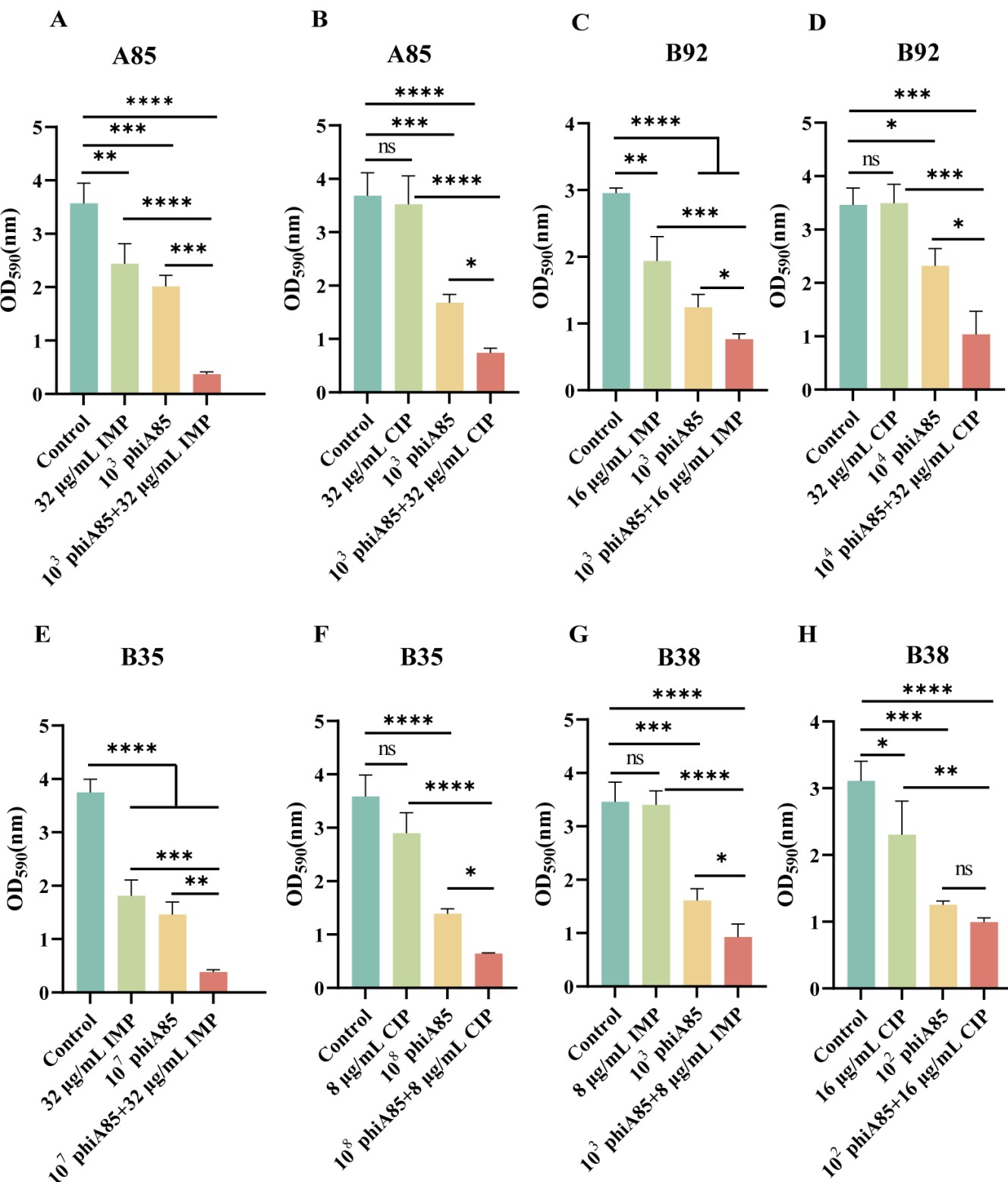

**FIG 6** The inhibitory effect of the phiA85 and antibiotic mixtures on biofilm formation. (A and B) Phage phiA85 and imipenem or ciprofloxacin mixtures on A85. (C and D) Phage phiA85 and imipenem or ciprofloxacin mixtures on B92. (E and F) Phage phiA85 and imipenem or ciprofloxacin mixtures on B35. (G and H) Phage phiA85 and imipenem or ciprofloxacin mixtures on B38. Data represent mean ± SD ($n = 3$). Statistical significance was analyzed by one-way analysis of variance with Tukey's multiple comparison test. *$P < 0.05$, **$P < 0.01$, ***$P < 0.001$, ****$P < 0.0001$. ns, not significant.

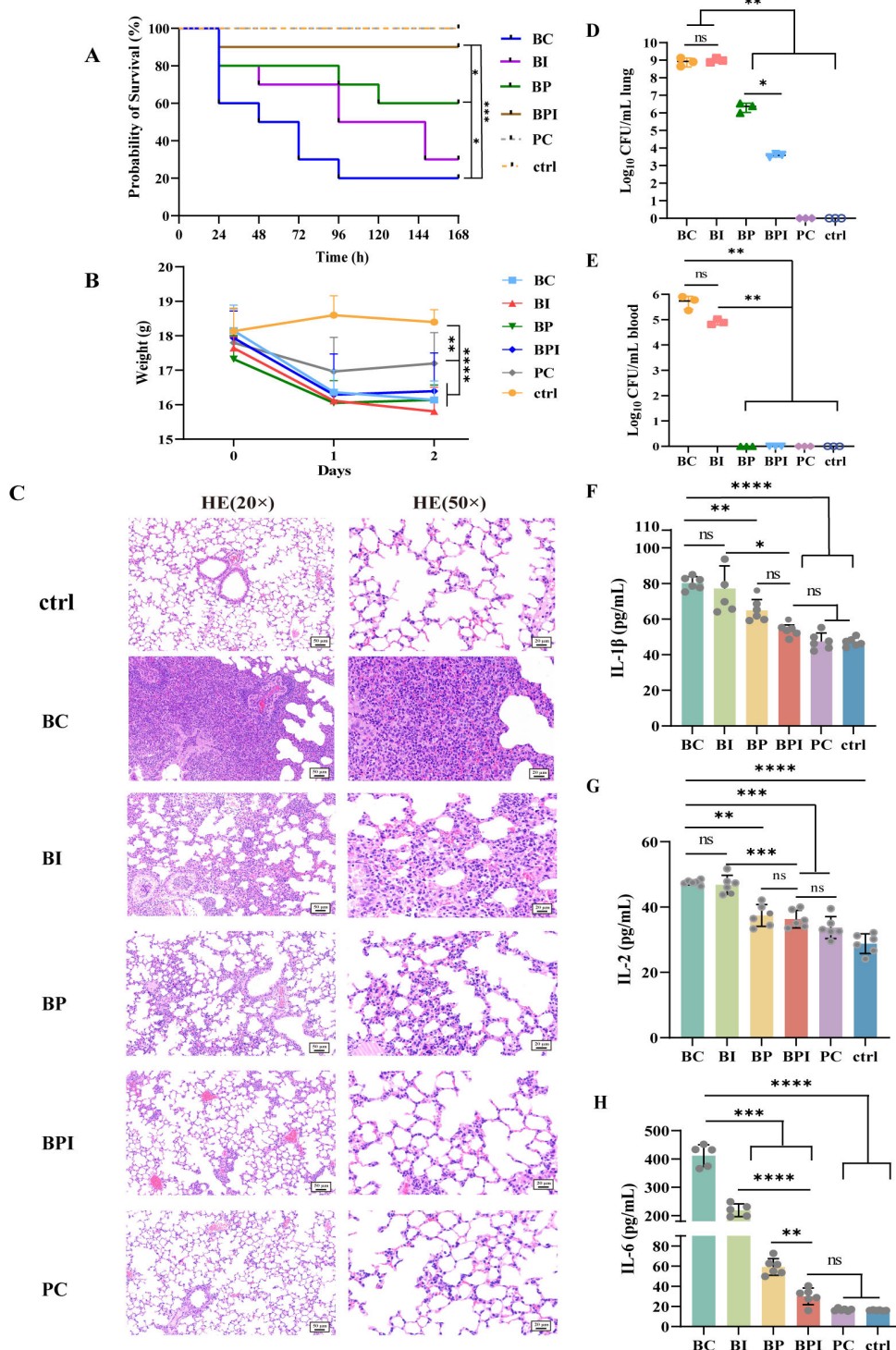

**FIG 7** The phage–antibiotic combination therapy in a mouse pneumonia model. ctrl, control group; BC, bacteria control; BI, bacteria + imipenem group; BP, bacteria + phage group; BPI, bacteria + phage–imipenem combination group; PC, phage only. (A) Survival rates of 6-week-old female BALB/c mice ($n = 10$) following intratracheal instillation. (B) The body weight of mice ($n = 6$) before and after infections (1 and 2 days). (C) Hematoxylin and eosin staining (×20 and ×50 magnifications) of lung sections. (D and E) The bacterial load of lungs and blood ($n = 3$) 24 h after infections. (F–H) The levels of serum cytokines (IL-1β, IL-2, and IL-6). Data represent mean ± SD ($n = 6$). Statistical significance was analyzed by one-way analysis of variance with Tukey's multiple comparison test. *$P < 0.05$, **$P < 0.01$, ***$P < 0.001$, ****$P < 0.0001$. ns, not significant.

## The synergistic effect of phage phiA85–antibiotic combinations on biofilm formation

Biofilm formation, which is a critical determinant in the development of antibiotic resistance, plays an important role in the pathogenesis of bacterial infections (34, 35). The synergistic effect of phage phiA85–antibiotic combinations on biofilm formation was evaluated through crystal violet assay. The results showed that phage phiA85–antibiotic groups could effectively inhibit biofilm formation in comparison with antibiotics monotherapy groups (Fig. 6). Furthermore, phage phiA85–imipenem and phage phiA85–ciprofloxacin combinations in major groups better inhibited biofilm formation than phage monotherapy groups (Fig. 6), which indicated that phage–antibiotic combinations had a synergistic effect in biofilm inhibition compared with phage phiA85 treatment alone.

## Phage phiA85–imipenem combination can effectively treat mouse pneumonia

To assess the therapeutic effect of phage phiA85–imipenem combination *in vivo*, a mouse pneumonia model was utilized through intratracheal instillation of A85. The survival curves indicated that intratracheal instillation of A85 can cause 80% mortality within 7 days. Phage phiA85–imipenem combination effectively reduced mice mortality, compared to monotherapy (Fig. 7A). Compared with the control group, mice experiencing bacterial infection showed a decrease in body weight, reflecting the decrease in appetite in the infected state regardless of treatment (Fig. 7B). The lung tissues infected with A85 showed congestion and necrosis (Fig. S3B). Hematoxylin and eosin (HE) staining exhibited disruption of normal alveolar structure and diffuse neutrophil infiltration. The degrees of lung lesions were reduced in phage phiA85 treatment. HE staining showed a thickening of the alveolar wall and slight neutrophil infiltration. Furthermore, the therapeutic effects of the phage–imipenem combination were more significant, indicating a more complete alveolar structure and fewer inflammatory cells in pathological staining (Fig. 7C). In addition, we explored the bacterial load in lungs and blood of mice 24 h post-infection. As expected, the bacterial load of lungs in bacteria control and imipenem monotherapy exceeded the initial bacterial dose (Fig. 7D). Bacteria were also isolated in the blood of these two groups, which indicated that the bacteria invaded the bloodstream while replicating in the lungs, causing bacteremia in mice (Fig. 7E). In phage treatment and phage–imipenem combination treatment groups, the bacterial load of lungs was less than the initial, and bacteria were not isolated in blood samples (Fig. 7E). Meanwhile, compared to the control group, serum cytokines (interleukin [IL]-1β, IL-2, and IL-6) in mice infected with A85 were significantly elevated, suggesting a systemic inflammatory response. Serum cytokine levels reduced in phage treatment and phage–imipenem combination treatment groups compared to the bacterial infection (bacteria control [BC]) group. However, except for serum levels of IL-6, no significant differences were observed in serum cytokine levels between the phage-treated and phage–imipenem combination groups, which may indicate that the systemic inflammatory response was not so significantly different between the phage-treated and phage–antibiotic combination groups of mice (Fig. 7F through H). These results indicated that phage phiA85–imipenem combination therapy was more effective than phage monotherapy in treating A85-induced mouse pneumonia.

## DISCUSSION

According to the released updated Bacterial Priority Pathogens List 2024 from the World Health Organization, third-generation cephalosporin-resistant enterobacterales and carbapenem-resistant enterobacterales are listed as the critical categories for prioritization. The list provides guidance on the development of new and necessary treatments to stop the spread of antimicrobial resistance (36). Several MDR-CRKP strains mentioned in this study belong to these two critical categories of pathogens mentioned

above. A85, which we isolated from the clinic, carries typical hypervirulence genes of *K. pneumoniae* as well. Therefore, it is of significant importance to investigate novel therapeutic strategies as alternatives to conventional antibiotics for CRKP infections. Phages are emerging as a promising tool in the era of decreasing effective antibiotics. The advantages of phage therapy over antibiotics are its host specificity, non-disruption of normal flora, and lower developmental costs relative to developing new drugs (37). In our study, phage phiA85 exhibited effectively lytic activity and biofilm inhibition on several CRKP and other host strains.

Recent studies have extensively reported *Klebsiella* phages, with the majority of these phages demonstrating lytic activity exclusively against hosts with specific KL types (38–40). Phages enter the host cell by producing depolymerase, which is capable of recognizing, binding, and digesting specifically oligosaccharide bonds of capsule (41). Consequently, the diversity of KL types in *K. pneumoniae* restricts the host range of *Klebsiella* phages. Several studies have also reported about the broad host range *Klebsiella* phages. For instance, Pan et al. identified a phage capable of infecting 10 different KL types of *K. pneumoniae* strains and characterized nine functional capsular depolymerase-encoding genes (25). Based on phages isolated from strains of diverse capsular types, researchers have designed broad host range phage cocktails, comprising 12 phages, that demonstrated activity against 55% of reference *Klebsiella* serotypes, with 31% of CRKP strains showing susceptibility (42). A detailed study comparing narrow-range (capsular-specific) and broad-range phages revealed distinct phage–host interaction mechanisms. Unlike narrow-host-range phages encoding depolymerases, broad-host-range phages lack canonical depolymerase genes, suggesting alternative entry strategies (43). This aligns with the characteristics of our phage phiA85. Combined with the absence of plaque haloes and no homology sequence alignments, it is initially concluded that our isolated phage phiA85 does not encode canonical depolymerase. This suggests that, unlike those narrow-host-range phages that encode depolymerases, phage phiA85 harbors additional receptor binding proteins, which lead to alternative modes of host-phage recognition independent of bacterial KL types. Through screening for phiA85-resistant mutants, we isolated a strain and identified a point mutation in the outer membrane protein OmpC through genome sequencing (Fig. S4). No genetic alterations were detected within the capsular synthesis cluster. These findings preliminarily suggested that the host-phage interaction involving phiA85 might be associated with outer membrane proteins. However, the current evidence remains insufficient to substantiate this hypothesis. In this study, phage phiA85 was able to lyse host strains of different KL types and O types. The repeating units of polysaccharides exhibit structural variations across distinct O antigens or KL types (44). Therefore, we speculated that phages cannot enter host strains through binding to structurally conserved motifs. In addition, the uncharacterized functions of multiple hypothetical proteins encoded by phage phiA85 warrant further in-depth investigation in future studies. The mechanism of phage phiA85 and host interaction needs further investigation. Lourenço et al. reported a broad-host-range phage, mtp5, targeting non-capsulated *K. pneumoniae*, which was able to lyse multiple genetic sublineages and O types of host strains. Moreover, this phage induces a lower rate of resistance emergence *in vitro*. The study also revealed that mutations associated with resistance against the broad-range mtp5 were linked to the O-antigen locus and lipopolysaccharide biosynthesis genes and partially involved the errichrome-iron outer membrane transporter as well as carbohydrate metabolism-related genes through sequencing-induced phage-resistant strains (26). This provides insights for identifying bacterial receptors of broad-host-range phages. In this study, the host-range test showed that phage phiA85 was able to lyse MDR-CRKP of different KL types (KL24, KL27, KL47, KL109, and KL136). This feature also expands the therapeutic applicability of phiA85, making them particularly effective for multiple bacterial infections (45). Whereas phage phiA85 is not restricted by KL types, it retains strict species specificity, selectively lysing *K. pneumoniae* instead of other bacterial species.

Our study further demonstrated that the phage phiA85–antibiotic combination had a synergistic effect compared to monotherapy against CRKP *in vitro* and *in vivo*. Research on the mechanism of PAS began in 2007, with studies reporting that sublethal concentrations of β-lactam and quinolone antibiotics inhibit bacterial division and cause bacterial filamentation, leading to an increase in phage assembly (46). It has also been shown that the prolongation of the phage assembly period in PAS is due to delayed host lysis, as the increase in bacterial surface area outweighs the production of phage holin, a phage protein that makes holes in inner membrane and promotes bacterial lysis. Reactive oxygen species stress also led to an increased production of phages (47). In addition, there is another possibility that sub-MIC concentrations of antibiotics may activate and release the prophages within bacteria, thereby exacerbating bacterial death (48). In our study, phage–imipenem or ciprofloxacin combinations exhibited better bactericidal effect compared with monotherapy. We speculated that the mechanism of action might be that under the subinhibitory concentrations of imipenem or ciprofloxacin, the poor bacterial division and the weakened cell wall increased the biosynthesis and assembly of phage. More research is needed to explore the mechanism of PAS. The administration of phage exerts substantial selective pressure on bacterial populations, thereby driving the emergence of phage-resistant variants. There is growing evidence that phage–antibiotic combinations are more effective in controlling pathogens while reducing the emergence rate of phage-resistant bacteria as opposed to monotherapy (29, 49, 50). Consequently, the phage–antibiotic combination holds great therapeutic potential for patients with clinically multidrug-resistant infections who have failed antibiotic therapy. However, the selection of antibiotic, as well as the dosage of antibiotics and phage, needs to be carefully adjusted, guided by *in vitro* experiments and actual situations of the clinical patients. In conclusion, our study characterized a broad-host-range phage phiA85 with general lytic activity and biofilm inhibition effect against different KL types and ST types of strains. We also demonstrated that the phage phiA85–antibiotic combinations had a synergistic effect against MDR-CRKP compared to phage monotherapy on *in vitro* experiments and mouse pneumonia models. These findings provide a promising option for the treatment of MDR-CRKP infections.

## MATERIALS AND METHODS

### Bacterial isolates and growth conditions

All strains were isolated from clinical samples in Capital Institute of Pediatrics, Beijing, China. Bacteria were incubated in LB broth (5 g/L yeast extract, 10 g/L sodium chloride, and 10 g/L tryptone) in a shaking incubator (180–200 rpm) at 37°C. A MDR-CRKP strain (A85) was used as a host for the isolation and proliferation of phage. The genomic DNA of the above strains was extracted and sequenced using the Illumina Hiseq platform by Novogene. All the resistant genes and hypervirulence genes were annotated in https://bigsdb.pasteur.fr/klebsiella/.

### String test

*K. pneumoniae* isolates were cultured overnight on LB agar plate at 37°C. An inoculation loop touched the surface of a single colony. The positive string test showed that a single colony stretched with an inoculation loop can form a viscous string >5 mm in length.

### Mucoviscosity test

The bacterial culture was incubated overnight and centrifuged at 2,000 rpm for 5 min. The supernatant was removed without disturbing the pellet. $OD_{600}$ measurement was

performed to determine the bacterial mucoviscosity. Three independent cultures were used for each assay.

## Phage isolation, purification, and amplification

The phage was isolated from sewage water collected from Capital Institute of Pediatrics, Beijing, China. Briefly, 100 µL A85 in logarithmic phase was mixed with 100 µL filtered hospital sewage in 2 mL LB broth. The mixture was incubated overnight at 37°C in 200 rpm shaking condition and then centrifuged at 12,000 rpm for 10 min. The supernatant was obtained after filtering. Spot tests were used to confirm the lytic effect of phage in the supernatant. A clear plaque was considered as positive bacteriolytic activity. Individual plaques were selected for phage purification using the double-layer agar method (51) three times until homogeneous plaques were formed on a double-layer agar plate.

## Optimal multiplicity of infection and one-step growth curve

Different phage concentrations were mixed with the logarithmic phase of host strains in LB broth, and the optimal MOI was measured through a double-layer plaque assay. The host strain in logarithmic phase was mixed with the phage at the optimal MOI and incubated at 37°C in 20 mL LB broth. The phage titers were calculated every 10 min for a total duration of 150 min to generate a one-step growth curve.

## Phage stability evaluation

About $10^8$ PFU phages were incubated at 25°C, 37°C, 50°C, 60°C, 70°C, 80°C, 90°C, and 100°C and pH from 3 to 13 (phage suspension was diluted in sodium-magnesium buffer at varying pH levels, adjusted using HCl and NaOH solutions) for 1 h, respectively. Next, the phage titers were identified by using a double-layer plaque assay.

## Transmission electron microscopy

Phage particles were negatively stained with 2% (wt/vol) uranium acetate (pH 7) and observed by TEM operated at 80 kV.

## Genome sequencing and analysis

The genomic DNA of the phage phiA85 was extracted by the phenol-chloroform method as previously described (52). Whole-genome sequencing of DNA was performed on the Illumina HiSeq 2500 platform (Berry Genomics Corp., China). *De novo* assembly of clean reads was conducted using the metaSPAdes assembler (53), with systematic *k*-mer length variation testing performed to obtain the optimal assembly outcome. Potential CDSs were annotated using prokka 1.14.6 (54). Putative homologies with phage-related proteins were detected through blastp alignment. The Proksee platform (https://proksee.ca/) (55) was employed for genome assembly and graphical construction. A phylogenetic tree was constructed using MEGA (version 11.0.13) software using the maximum-likelihood method and 1,000 bootstrap replications, based on amino acid sequences of large terminase subunit. Virulence-associated determinants were characterized through the VFDB database (56), whereas antibiotic resistance profiling was performed with Resfinder 4.0 (57). The phage's lifestyle (lytic/lysogenic) was predicted with the bacPHILP bioinformatics tool (58).

## Phylogenetic tree of host strains

A phylogenetic tree was constructed using the single-copy core genes identified through core-pan analysis. Single-copy core genes were identified by clustering the protein sequences of multiple analyzed samples using the CD-HIT software. The clustering threshold was set at 70% identity at the protein level. Protein sequences of these genes were aligned using MUSCLE software. The resulting multiple sequence alignments were

then used for tree construction. Neighbor-joining trees were constructed using TreeBeST. Bootstrap analysis with 1,000 replicates was performed to assess node support. The branch lengths represent the magnitude of evolutionary distance.

## Host range of the phiA85

The host range of the phage phiA85 was examined against 56 clinical *K. pneumoniae* isolates using a standard spot test and efficiency of plating (EOP) assay (59). The phage in 10 µL of LB broth was spotted onto the agar plate of host strains and incubated overnight. The appearance of clear and transparent plaques indicated susceptible strains, whereas the absence of plaques represented phage-immune strains. EOP values were calculated relative to the host strain A85.

## Screening of phage-resistant bacterial strains

One hundred microliters of bacteria in the logarithmic phase and 100 µL of phage capable of lysing the strains were mixed with 4 mL of semi-solid medium and then poured onto the surface of LB agar plates. The plates were incubated at 37°C until colonies appeared. Single colonies were then inoculated, and the sensitivity to phage infection was tested using phage plaque assays. Strains on which no plaques formed were selected as potential phage-resistant mutants.

## Antibiotic sensitivity test

The MICs of antimicrobial agents were determined using broth the microdilution susceptibility method and interpreted according to the Clinical and Laboratory Standards Institute guidelines (60).

## Killing curves

The lytic activity of the phages toward the host strains was tested at an MOI of 0.001, 0.1, and 10.0. Then, the $OD_{600}$ values were obtained on a microplate reader once an hour for each group.

## Synergy testing in LB broth

The synergy testing method was applied with slight modifications. The MDR-CRKP was cultured overnight and inoculated into 3 mL LB broth for 2 h. The subculture was centrifuged at 12,000 rpm for 10 min and washed with fresh LB media. Later, the bacteria were diluted to $1 \times 10^8$ CFU/mL and inoculated with 100 µL into each well of the 96-well plate. Varying concentrations of phage (50 µL, final $10^1$–$10^9$ PFU/mL) and antibiotic (50 µL, final 1–1,024 µg/mL) were gradient diluted and coated into a 96-well plate for a final $5 \times 10^7$ CFU/mL of bacteria per well. The $OD_{600}$ was measured once an hour for a total of 24 h. Phage–antibiotic killing dynamic curves were measured as killing curves. The concentrations of phage and antibiotics were mixed with bacteria in a 96-well plate; $OD_{600}$ was obtained on a microplate reader in the shaking mode once an hour for each group.

## Biofilm formation quantification

Strains were grown overnight, diluted to $OD_{600}$ of 0.01 in LB broth, and incubated at 100 µL with 100 µL phage phiA85 or antibiotics in each well of a 96-well plate at 37°C for 24 h. Subsequently, the liquid was poured from the medium. Each well was washed with 1× phosphate-buffered saline (PBS) and stained with 1% crystal violet for 30 min. Afterward, the excess crystal violet was removed and washed twice with 1× PBS. Then, each well was eluted with 200 µL of a bleaching solution (methanol:glacial acetic acid:$H_2O$ in a 4:1:5 ratio). The plate was incubated with gentle shaking at room temperature for 30 min. Finally, biofilm quantification with crystal violet was measured by detecting the optical density at a wavelength of 590 nm.

## Mouse infection model

MDR-CRKP A85 cultures were grown in LB medium to an $OD_{600}$ of 1.0. The cultures were washed twice and resuspended in 1× PBS. Each mouse received intratracheal instillation with 50 µL of the bacterial suspension, delivering a total of $2 \times 10^8$ CFU. Intratracheal instillation infections were performed in 6-week-old female BALB/c mice. Mice were randomly divided into six groups: (i) a control group, which received intratracheal instillation of saline; (ii) a bacteria group (BC), which received intratracheal instillation of $2 \times 10^8$ CFU A85; (iii) a bacteria + imipenem group, which received intratracheal instillation of $2 \times 10^8$ CFU A85 and 32 µg imipenem 1 h later; (iv) abacteria + phage group, which received intratracheal instillation of $2 \times 10^8$ CFU A85 and $10^3$ PFU phiA85 1 h later; (v) a bacteria + phage–imipenem combination group, which received intratracheal instillation of $2 \times 10^8$ CFU A85 and $10^3$ PFU phiA85 with 32 µg imipenem mixture 1 h later; and (vi) a phage group, which received intratracheal instillation of $10^3$ PFU phiA85. Survival rates ($n = 10$) were monitored for 7 days post-infection. The body weights ($n = 6$) of mice were measured and recorded before treatment, 1 and 2 days after treatment, respectively. Bacteremia levels ($n = 3$) were assessed by plating CFUs from blood and lung samples. Serum samples ($n = 6$) were collected to determine cytokine levels (IL-1β, IL-2, and IL-6), and lung tissues were used for hematoxylin and eosin staining at 36 h post-infection.

## Statistical analysis

Data were presented as mean ± SD and analyzed by one-way analysis of variance with GraphPad Prism (version 9.5, USA). Significance comparisons between two groups of data were analyzed by Student's $t$-test. Survival rates were compared by the Kruskal–Wallis test. *$P < 0.05$, **$P < 0.01$, ***$P < 0.001$, and ****$P < 0.0001$ indicate levels of statistical significance. Each data were included in at least three biological replicates.

## ACKNOWLEDGMENTS

This research was funded by grants from the National Natural Science Foundation of China for Key Programs of China Grants (82130065) and the National Natural Science Foundation (32200159 and 32400157).

Conceptualization: J.Y., Z.L., and Z.F.; methodology: Z.L., Z.F., T.F., Y.C., H.L., X.C., B.D., G.X., Y.F., H.Z., J.C., C.Y., L.G., J.F., Z.X., and Z.Y.; writing (original draft): Z.L.; writing (review and editing): J.Y.

## AUTHOR AFFILIATIONS

[1]Department of Bacteriology, Capital Institute of Pediatrics, Beijing, China
[2]Capital Institute of Pediatrics, Chinese Academy of Medical Sciences & Peking Union Medical College, Beijing, China

## AUTHOR ORCIDs

Jing Yuan ⓘ http://orcid.org/0000-0002-6939-9676

## FUNDING

| Funder | Grant(s) | Author(s) |
| --- | --- | --- |
| National Natural Science Foundation of China | 82130065 | Jing Yuan |
| National Natural Science Foundation of China | 32200159 | Zheng Fan |
| National Natural Science Foundation of China | 32400157 | Tongtong Fu |

## AUTHOR CONTRIBUTIONS

Zhoufei Li, Conceptualization, Methodology, Writing – original draft | Zheng Fan, Data curation, Supervision, Writing – review and editing | Tongtong Fu, Software, Validation | Yuchen Chen, Data curation, Methodology | Lin Gan, Investigation, Methodology | Bing Du, Formal analysis | Xiaohu Cui, Software | Guanhua Xue, Validation | Yanling Feng, Visualization | Hanqing Zhao, Project administration | Jinghua Cui, Supervision | Chao Yan, Data curation | Junxia Feng, Software | Ziying Xu, Data curation | Zihui Yu, Validation | Yang Yang, Visualization | Yuehua Ke, Software | Jing Yuan, Conceptualization, Funding acquisition, Writing – review and editing

## DATA AVAILABILITY

The complete genome sequences of phage phiA85 and its host bacterial *Klebsiella pneumoniae*
strains have been deposited in the National Center for Biotechnology Information GenBank database. The accession numbers are shown in Table S3.

## ETHICS APPROVAL

All animal experiments were approved by the Capital Institute of Pediatrics Animal Care and Use Committee on the Ethics of Animal Experiments (permission no. DWLL2024022) and were in accordance with US and Chinese national guidelines for the use of animals in research.

## ADDITIONAL FILES

The following material is available online.

### Supplemental Material

**Fig. S1 (Spectrum02019-25-s0001.tif).** The genomic map of the chromosome of A85 (A85 chr).
**Fig. S2 (Spectrum02019-25-s0002.tif).** A. Killing curves of phage phiA85 against host strains. B. The inhibitory effect of the phiA85 on biofilm formation of host strains.
**Fig. S3 (Spectrum02019-25-s0003.tif).** A. Killing curves of phage phiA85 against host strains. B. Anatomy images of mice lungs in different groups.
**Fig. S4 (Spectrum02019-25-s0004.tif).** A. The colony morphology of A85 (left) and its phage mutant strains (right). B. Schematic diagram of phage mutant mutation sites.
**Supplemental material (Spectrum02019-25-s0005.docx).** Supplemental figure legends.
**Table S1 (Spectrum02019-25-s0006.xlsx).** The host range of phage phiA85.
**Table S2 (Spectrum02019-25-s0007.xlsx).** The lifecycle prediction of phage phiA85.
**Table S3 (Spectrum02019-25-s0008.xlsx).** The GeneBank number of phage and bacteria.

### Open Peer Review

**PEER REVIEW HISTORY (review-history.pdf).** An accounting of the reviewer comments and feedback.

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
