## [Reviewer comments · Microbiology Spectrum]

Microbiology Spectrum

A novel broad host range phage phiA85 displays a synergistic effect with antibiotics targeting carbapenem-resistant *Klebsiella pneumoniae*

Zhoufei Li, Zheng Fan, Tongtong Fu, Yuchen Chen, Lin Gan, Bing Du, Xiaohu Cui, Guanhua Xue, Yanling Feng, Hanqing Zhao, Jinghua Cui, Chao Yan, Junxia Feng, Ziyang Xu, Zihui Yu, Yang Yang, Yuehua Ke, and Jing Yuan

Corresponding Author(s): Jing Yuan, Capital Institute of Pediatrics

Review Timeline:

Submission Date:	July 2, 2025
Editorial Decision:	July 14, 2025
Revision Received:	July 15, 2025
Accepted:	July 16, 2025

Editor: Estela Galvan

Reviewer(s): The reviewers have opted to remain anonymous.

Transaction Report:

DOI: <https://doi.org/10.1128/spectrum.02019-25>

Re: Spectrum02019-25 (**A novel broad host range phage phiA85 displays a synergistic effect with antibiotics targeting carbapenem-resistant *Klebsiella pneumoniae***)

Dear Prof. Jing Yuan:

Thank you for the privilege of reviewing your work. Below you will find my comments, instructions from the Spectrum editorial office, and the reviewer comments.

Revision Guidelines

Sincerely,
Estela Galvan
Editor
Microbiology Spectrum

I am pleased to inform you that your manuscript has been editorially accepted for publication. However, there are a few additional questions in the submission form that need to be answered before the final decision. Once these are completed, please return your submission so that I can move your paper forward to acceptance.

Point-to-Point Response

Dear Reviewer,

Thank you for your thorough review of my manuscript and response. We feel grateful for the time and effort you dedicated to evaluating my work. Your insightful and meaningful comments have significantly helped me improve the quality of this paper.

We have carefully addressed all your suggestions and revised the manuscript accordingly. Below, I provide a point-by-point response to your specific comments. Please let me know if any additional clarification or modification is needed.

Reviewer #2 (Comments for the Author):

The concerns raised in my previous review have been addressed.

However, the methodology for the construction of the phylogenetic tree should be detailed more. How were gene families identified? what was the threshold of identity? What is the scale of Figure 3B (phylogenetic tree)?

It is also a pity that authors don't re-order strains in Figure 3A according to the phylogenetic tree presented in Fig 3B.

Response :

Thanks for your considerable comments. The phylogenetic tree was constructed based on core-pan analysis. Single-copy core genes were identified by clustering the protein sequences of multiple analyzed samples using the CD-HIT software. Sequence similarity threshold: sequences with $\geq 70\%$ similarity will be clustered. Both alignment length and similarity must reach 70%. The protein multiple sequence alignment was performed using MUSCLE software, and subsequently, the phylogenetic tree was constructed. We have added the content in Methods (L412-418).

For visual clarity, the previous version of the phylogenetic tree image was one that did not consider branch lengths. We have now replaced it with a version that includes a scale bar (Figure 3B). The branch lengths represent the magnitude of evolutionary distance, calculated as the number of substitutions per nucleotide site. We have re-order the strains according to the phylogenetic tree in Figure 3A.

Editor comments:

Please arrange for release of any genome data deposited in public databases.

Response :

All genomic information for the bacteria has been released in the NCBI database, and the genebank numbers have been modified in the Supplemental Table 3.

We really appreciate the reviewers for their constructive feedback, which has significantly strengthened the manuscript. All required changes have been made in the article as required in the revised version.

Sincerely,

Jing Yuan

Re: Spectrum02019-25R1 (**A novel broad host range phage phiA85 displays a synergistic effect with antibiotics targeting carbapenem-resistant *Klebsiella pneumoniae***)

Dear Dr. Jing Yuan:

Your manuscript has been accepted, and I am forwarding it to the ASM production staff for publication. Your paper will first be checked to make sure all elements meet the technical requirements. ASM staff will contact you if anything needs to be revised before copyediting and production can begin. Otherwise, you will be notified when your proofs are ready to be viewed.

Sincerely,
Estela Galvan
Editor
Microbiology Spectrum